# Ecological and reproductive characteristics of holothuroids *Isostichopus badionotus* and *Isostichopus* sp. in Colombia

Ernesto J. Acosta[1,2,3,4]*, Adriana Rodríguez-Forero[5], Bernd Werding[4], Andreas Kunzmann[1]

**1** WG Ecophysiology & Experimental Aquaculture, Leibniz Centre for Tropical Marine Research (ZMT) GmbH, Bremen, Germany, **2** Corporation Center of Excellence in Marine Sciences CEMarin, Bogotá, Colombia, **3** Instituto de Investigaciones Marinas y Costeras INVEMAR, Santa Marta, Colombia, **4** Department of Animal Ecology, Justus-Liebig-University Giessen, Giessen, Germany, **5** Aquaculture Research and Technological Development Group, Aquaculture Laboratory Fisheries Engineering Program, Universidad del Magdalena Santa Marta, Santa Marta, Colombia

\* ernesto77.acosta@gmail.com

**Data Availability Statement:** The database for this manuscript is available at: https://osf.io/qtac7/.

## Abstract

*Isostichopus badionotus* and *Isostichopus* sp. are two holothuroids exploited in the Caribbean region of Colombia. Until recently, they were considered a single species. During one year, 222 individuals of *Isostichopus* sp. and 114 of *I. badionotus* were collected in two bays of the Santa Marta region to study their reproductive biology and collect information on their size, weight and habitat. Both sea cucumber morphotypes showed an annual reproductive cycle, with a reproductive season from September to November, closely related to the increase in water temperature and rainfall. In both sea cucumbers the population structure exhibited a unimodal distribution composed of mature individuals and a sex ratio of 1:1. *Isostichopus* sp. had an average size and weight (193 ± 52 mm and 178 ± 69 g) and size and weight at first maturity (175 mm and 155 g) that was much lower than *I. badionotus* (respectively, 324 ± 70 mm and 628 ± 179 g; 220 mm and 348 g). While 98% of *Isostichopus* sp. individuals were collected in the upper 2.5 m, on rocky bottoms between cracks, 73% of *I. badionotus* individuals were found between 3 and 7.8 m depth, exposed on sandy bottoms. These differences imply that management measures (e.g. minimum catch size) should not be the same for both sea cucumbers morphotypes.

## 1. Introduction

Although sea cucumber fishing is not a traditional activity in Colombia, for almost two decades it has become an additional source of income for indigenous communities and fishermen on the Colombian Caribbean coast, who extract this resource with practically no regulation [1–4]. Fishing experience in neighboring countries, such as Panama, Mexico and Ecuador have shown the vulnerability of sea cucumbers to over-exploitation due to low recruitment rates, late maturity and density-dependent reproduction [5–7]. However, in Colombia this

**Funding:** E.A. This study was funded by corporation center of excellence in marine sciences CEMarin Call N° 5 of year 2015. https://www.cemarin.org The funders had no role in study design, data collection and analysis, decision to publish, or preparation of the manuscript.

**Competing interests:** The authors have declared that no competing interests exist.

activity continues to be carried out illegally on different species and tends to be mostly unregulated and unquantified [8], encouraged by growing demand from Asian markets, where sea cucumber is considered a gastronomic delight of high economic value [8–10].

Existing records on this fishery as well as confiscations made by Colombian environmental authorities indicate that one of the species most frequently caught and of greatest concern for its protection and conservation is *Isostichopus badionotus* [1, 3, 11], because it is one of the most valuable in the Caribbean reaching prices in Asian markets between 132 and 358 USD Kg$^{-1}$ [12]. Sadly, fishermen who sell sea cucumbers to intermediaries receive an average of only 3 USD Kg$^{-1}$ [1, 3].

Despite the fact that *I. badionotus* has several morphotypes with a wide variation in its coloration and external morphology, this species has been recognized as a single species based on ossicle morphology, which is the main taxonomic character in sea cucumbers [13, 14]. However, recently Vergara and et al. [15], based on DNA analysis, morphology, and habitat preferences, found that one of these morphotypes shows a different genetic lineage and may not correspond to the *Isostichopus badionotus* species. They provisionally called it *Isostichopus* sp. and for practical reasons in this work we will call it morphotype *Isostichopus* sp., while the other morphotype is called *I. badionotus*.

Notwithstanding the fishing pressure on the different morphotypes of *I. badionotus*, as well as on other species of sea cucumbers, research on these exploited species in Colombia has been limited to a few studies on abundance and distribution [2, 16] and reproductive biology [17, 18]. However, there are still many gaps in knowledge, particularly about their reproductive patterns and how they are influenced by variables such as temperature, salinity, rainfall, and food availability.

The present study describes aspects of the reproductive biology of the holothuroids *I. badionotus* and *Isostichopus* sp. in the Santa Marta Region using the gonadal index (GI) and histological examination of the gonads. It also evaluates the influence of environmental and chemical variables (temperature, rainfall, the percentage of organic carbon content in the sediment) on their reproductive biology and analyzes the structure of their populations (frequency of length and weight, sex ratio, mean size and weight at first sexual maturity, fecundity). This research is intended to increase the current knowledge about the biology and ecology of sea cucumbers native to the Colombian Caribbean Sea and provide a scientific basis for the development of management measures and conservation of this resource.

## 2. Materials and methods

Specimens of *Isostichopus* sp. were collected from December 2015 to January 2017, while specimens of *Isostichopus badionotus* were collected from February 2016 to January 2017 in the Taganga Bay (11˚ 16' 07.42" N– 74˚ 11' 37.16" W) and Rodadero Bay (11˚ 12' 27.40" N– 74˚ 13' 47.44" W) in the Caribbean region of Santa Marta, Colombia (Fig 1).

Due to the wide variety of shapes and colors that *I. badionotus* has, this study focused on the collection of organisms with the most common morphotype of this species present in the study area, described by Vergara et al. [15] as *I. badionotus* morphotype II.

Monthly 14 to 20 individuals of each morphotype (*I. badionotus* and *Isostichopus* sp.) were collected by snorkeling. For this purpose, a pair of snorkelers swam a transect 5 m wide by approximately 1000 m long, twice. The first trip was made bordering the coast line and the second at a distance of approximately 10 m from the coast line. The maximum search depth was 10 m for logistical and safety reasons and although it is likely that there are individuals at greater distance and depth, this could not be assessed in the present study. In October, no collections could be made due to the passage of Hurricane Matthew, which affected the Colombian Caribbean during that month. All sea cucumbers were individually packed in plastic bags

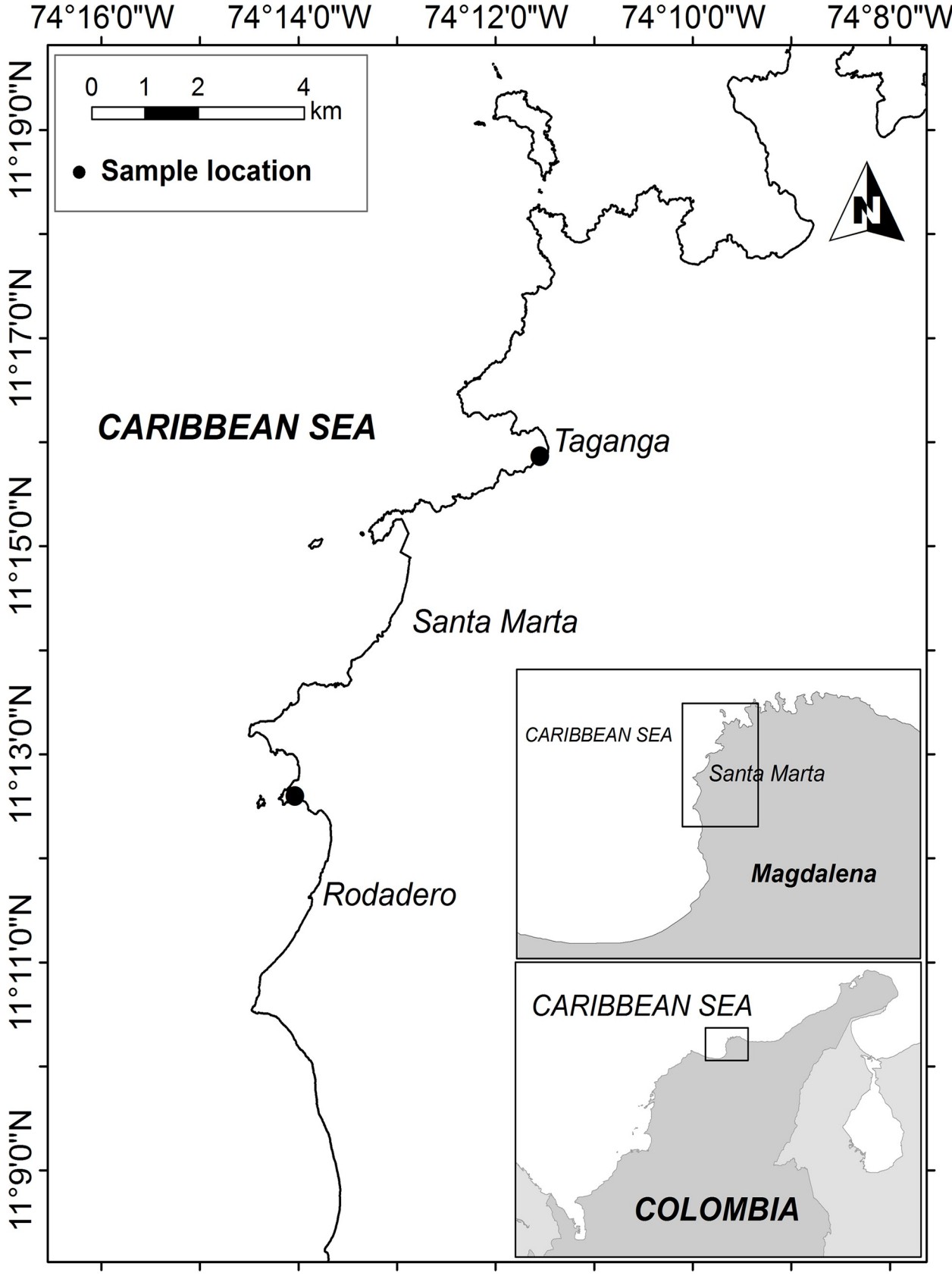

**Fig 1. Study area.** Box in inset A is magnified in inset B; box in inset B is magnified in the main panel. Source: Information base INVEMAR.

with seawater and transported to the Aquaculture Laboratory of Universidad del Magdalena. The depth of capture of each individual and the type of bottom it occupied were recorded (classified as rocky, sandy, coral, rock-coral, rock-sand, rock-sand-coral). Additionally, surface sediment samples (with a thickness not greater than 10 mm) were taken near the individuals to determine the percentage of organic carbon content using a mass spectrometer, Euro EA 3000 elemental analyzer (EuroVector). Salinity and water temperature were recorded on a monthly basis, while total rainfall data were taken from the records of the Simon Bolivar airport meteorological station of the city of Santa Marta.

The individuals were sacrificed in the laboratory by placing them in seawater at 4°C, which was also intended to reach a maximum state of contraction that allowed to reduce the error during length measurements, due to the capacity that they have to extend or contract at will. Then the following body measurements were taken: total body length (TL), using a flexible 1-m tape measure at the nearest 0.5 cm (after sacrifice by thermal shock the contracted body had a C or S shape, and the tape contoured the body, so that the total length is the contour length); gonad volume using a graduated cylinder at the nearest 0.1 cm$^3$; body wall weight (BW), defined as the weight of the individual without viscera or gonad and gonad weight (GW) at the nearest 0.1 g. Gonad index (GI) was calculated using the formula, GI = (GW/ BW)* 100 [19, 20].

The histological analysis of the gonads was performed at the Marine and Coastal Research Institute INVEMAR. A sample of approximately 2 g was taken from each of the gonads and processed in a Shandon Citadel 2000 automatic tissue processor according to the technique of dehydration, clarification and impregnation. The gonad tissue was cut to 5 μm thickness using a Microm HM32 manual rotary microtome and the sections were stained with hematoxylin-eosin [21].

Sex, maturity stage and occurrence of reproductive events were assigned using the classification scale of Ramofafia et al. [6] based on shape, color criteria and microscopic characteristics of oocytes and sperm: (I) Indeterminate, (II) growth, (III) mature, (IV) partially spawned and (V) spawned. Finally, the mean size at first sexual maturity (L50) and mean weight at first sexual maturity (W50) were defined as the length and weight at which the gonads of 50% of individuals were mature or had spawned (stages III, IV and V) [22].

To estimate the L50, the proportion of mature individuals at different size intervals was modelled using the following equation:

$$P = 1/(1 + e^{(a+b*TL)}),$$

where P = percentage of sexually mature females, TL = total length (mm), a and b = constants. The same equation was used to estimate the W50, replacing TL by BW = body wall weight.

Fecundity was estimated using the method described by Conand [22], starting from the average diameter of a mature oocyte (stage III), its volume was calculated assuming that it has a spherical shape and subsequently fecundity was estimated by dividing the average volume of the gonad of individuals in stage III by the average volume of an oocyte. The collection of individuals was carried out under the framework permit of the University of Magdalena registration number 1293 of December 18, 2013, granted by the national environmental licensing authority ANLA. Likewise, the processing of the specimens followed the international, national and institutional guidelines applicable to the care and use of animals.

## 3. Statistical analysis

The relationship between the sex of the individuals and the variables total length, body wall weight and gonad index were evaluated through one-way ANOVAS. The same analysis was

applied to compare the percentage of organic carbon in the nearby sediment of both sea cucumber morphotypes. Differences between treatments means were tested for significance using a post-hoc multiple comparison test (Tukey's HSD). Prior to data analysis, assumptions of normality were tested by the Kolmogorov-Smirnov test and homogeneity of variance by the Levene test. When the assumptions were not met, the data were appropriately transformed or the non-parametric Kruskal Wallis test was used in conjunction with the Bonferroni test. A regression analysis was performed with the monthly average of these variables to determine the relationship between environmental parameters (temperature, salinity and rainfall) and the GI Then a Spearman correlation analysis was performed to measure their strength of association. The sex ratio was evaluated with the Chi-square test. These analyses were carried out with the statistical program Statgraphics XVII.

The calculation of the L50 and W50 confidence intervals as well as their graphing was performed with the statistical program R version 3.6.3.

## 4. Results

### 4.1 Habitat description

*Isostichopus* sp. was found mostly in shallow waters, collecting 98% of the individuals in the first 2.5 m of depth and 2% between 2.6 and 4 m, generally hidden under rocks or in crevices, where 2 or 3 individuals shared the same refuge. Regarding bottom type, 94.5% of the individuals inhabited rocky bottoms, 4% rock-sand, 1% sandy, and 0.5% rock-coral.

*Isostichopus badionotus* was generally found at greater depths, 20% between 1 and 3 m, 73% between 3 and 8 m, and 7% between 8 and 10 m and in most cases the individuals were exposed. About 39% inhabited sandy bottoms with large rocks (diameter > 0.5 m), 27% sandy, 21% rock-coral, 8% rock-sand-coral and 5% rocky bottoms. No significant differences were observed between morphotypes in relation to the percentage of organic carbon in the sediment (*Isostichopus* sp. = 0.33% and *I. badionotus* 0.34%) ($n = 31$; $F_{1,29} = 0.24$; $p = 0.629$).

### 4.2 Population structure

From 222 individuals collected of *Isostichopus* sp., 91 (41%) were females, 68 (30.5%) males, 1 (0.5%) hermaphrodite and 62 (28%) did not have gonads. In *I. badionotus* from 114 individuals collected, 44 (39%) were females, 56 (49%) males, 14 (12%) did not have gonads. In both morphotypes, individuals without a gonad were observed in the months after the breeding season. The sex ratio was not significantly different from 1:1 (*I. badionotus*: $n = 100$; $X^2 = 1.4$; $p > 0.05$; *Isostichopus* sp.: $n = 158$; $X^2 = 3.3$; $p > 0.05$).

### 4.3 Body length-weight, distribution of size and weight and first maturity

**a) *Isostichopus* sp.** The average TL was 193 ± 52 mm (± SD), maximum 380 mm and minimum 75 mm. These values varied depending on sex (♀ = 195 ± 45 mm, ♂ = 202 ± 52 mm and without gonad 179 ± 61 mm). The individuals without a gonad were significantly smaller than the males ($n = 221$; $K = 6.513$; $p = 0.039$).

Length-frequency distribution was unimodal with a peak in the range of 180 to 215 mm (Fig 2A). The L50 was established at 175 mm (95% confidence interval: 174–176 mm) (Fig 2C). The smallest individual with gonads was a male (110 mm).

The average BW (± SD) was 179 ± 69 g, maximum 558 g and minimum 18 g (♀ = 187 ± 58 g, ♂ = 190 ± 71 g and without gonad 154 ± 77 g). The individuals without gonads were significantly less heavy than females and males ($n = 221$; $K = 11.410$; $p = 0.003$).

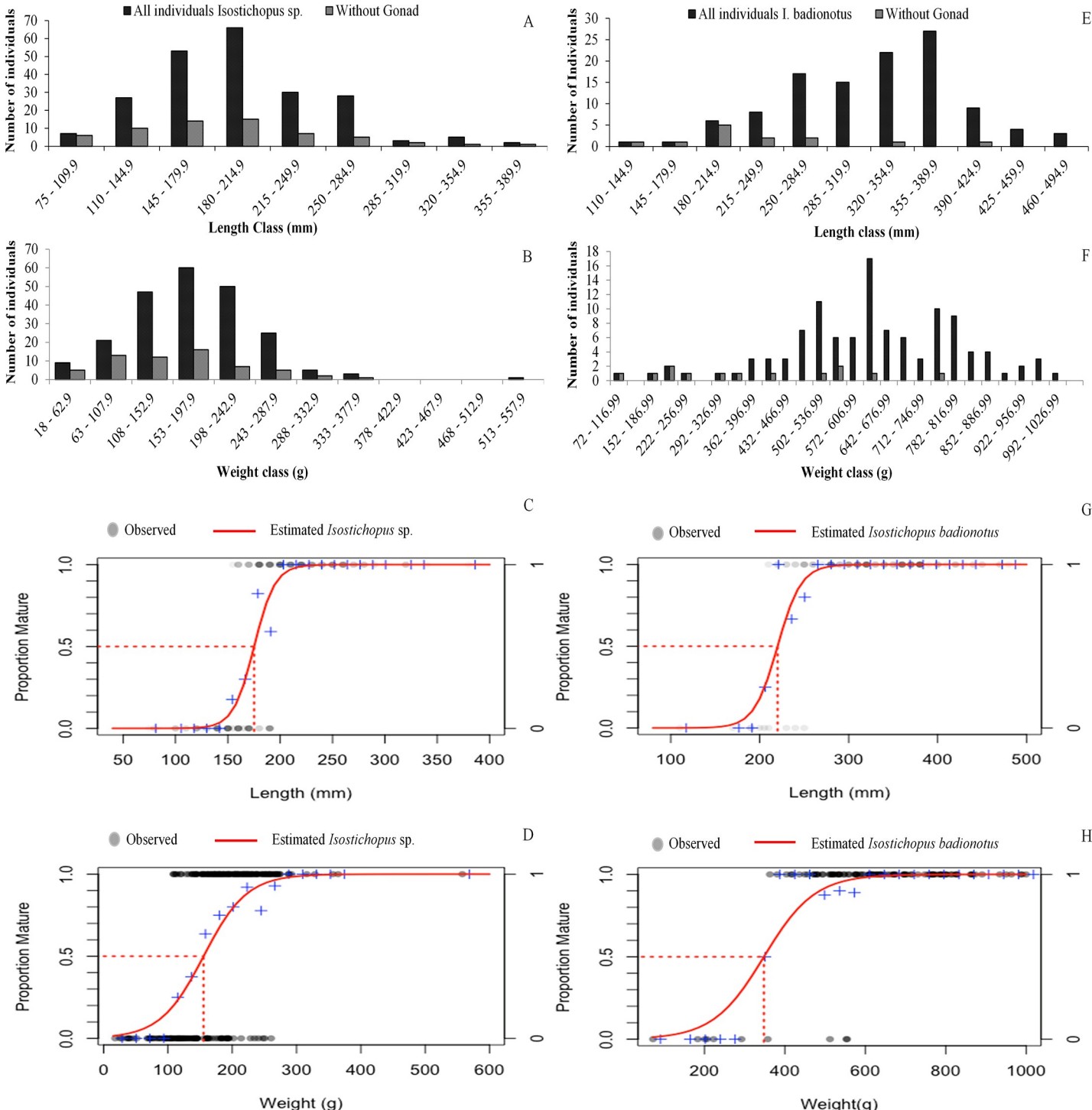

**Fig 2. Population structure.** A and B distribution frequency of length and weight *Isostichopus* sp.; E and F distribution frequency of length and weight *Isostichopus badionotus*; C and D length and weight at first maturity *Isostichopus* sp.; G and H length and weight at first maturity *Isostichopus badionotus*.

Frequency distribution of the BW was unimodal with a peak in the range of 153 to 198 g (Fig 2B). The W50 was established at 155 g (95% confidence interval: 154–156 g) (Fig 2D). The smallest individual collected with gonads was a male (39 g).

b) *Isostichopus badionotus.*   The average TL was 324 ± 70 mm (± SD), maximum 480 mm and minimum 110 mm (♀ = 329 ± 55mm, ♂ = 341 ± 64 mm and without gonad 240 ± 78 mm). The individuals without gonad were significantly smaller ($n$ = 114; $F_{2,111}$ = 14.74; $p$ = 0.001).

The length-frequency distribution was unimodal, with a peak in the range of 355 to 390 mm (Fig 2E). The L50 was established at 220 mm (95% confidence interval: 217–221 mm) (Fig 2G). The smallest individual collected with gonads was a male (210 mm).

The average BW was 628 ± 179 g, maximum 999 g and minimum 72 g (♀ = 654 ± 152 g, ♂ = 669 ± 145 g and without gonad 383 ± 200 g). The individuals without a gonad were significantly less heavy ($n$ = 114; $F_{2,111}$ = 20.14; $p$ = 0.001).

Frequency distribution of BW was unimodal with a peak in the range of 607 to 642 g (Fig 2F). The W50 was established at 348 g (95% confidence interval: 344–350 g) (Fig 2H). The smallest individual collected with gonads was a female (363 g).

## 4.4 Histological and morphological description of the gonads

In both sea cucumbers the gonad was composed of two tufts of tubules with a racemose appearance located at the anterior end of the coelom, which came to occupy most of the perivisceral cavity at the stage of maturity (III). The color of the gonad of both sea cucumbers varied from beige–whitish to ocher, without an apparent relationship between color and sex. Only one case of hermaphroditism was observed in *Isostichopus* sp., where 3% of the gonad contained sperm and 97% oocytes.

Histology showed 4 of the 5 stages of gonadal development proposed by Ramofafia et al. [6]. In this study the indeterminate stage (I), where the gametes do not yet have defined cellular structures that allow their differentiation, was not observed. A description of the histological characteristics of gonad development is shown in the S1 Appendix.

## 4.5 Gonad Index (GI)

In both morphotypes the lowest GI values were recorded in the first half of the year, with the lowest values in January 2016 for *Isostichopus* sp. (0.4 ± 0.2%) and February 2016 for *I. badionotus* (0.8 ± 0.6%). From May onwards, a gradual increase was observed, reaching its maximum value in August with 4.7 ± 0.9% and 7.6 ± 0.9%, respectively (Fig 3A and 3B), coinciding with the increase in water temperature and the onset of the rainy season (Fig 3C and 3D). Between September and December the GI decreased rapidly, indicating massive spawning, recording in January 2017 a GI of 1.5 ± 0.5% for *Isostichopus* sp. and 1.3 ± 0.4% for *I. badionotus*.

Both *Isostichopus* sp. and *I. badionotus* showed a significant relationship between GI and rainfall that explains more than 50% of the annual variation of GI. Temperature explained 24% and 42% of the annual variation of the GI, respectively. Finally, salinity explained 10% of the variation of the GI and did not show a significant influence (Table 1).

In general, the females had heavier gonads (Table 2) and a higher GI (Fig 3A and 3B), however, the differences in GI between females and males were only significant in *Isostichopus* sp. ($n$ = 159; $F_{1,157}$ = 18.39; $p$ = 0.001).

## 4.6 Reproductive cycle

*Isostichopus* sp. had by December 2015, 46.7% of mature individuals (III), 6.7% growth stage (II), 13.3% partially spawned (IV) and 33.3% without gonad. In January and February 2016, the percentage of individuals spawned (V), partially-spawned (IV) and without gonads increased (Fig 4A), indicating spawning activity, consistent with the GI decrease (Fig 3A). In

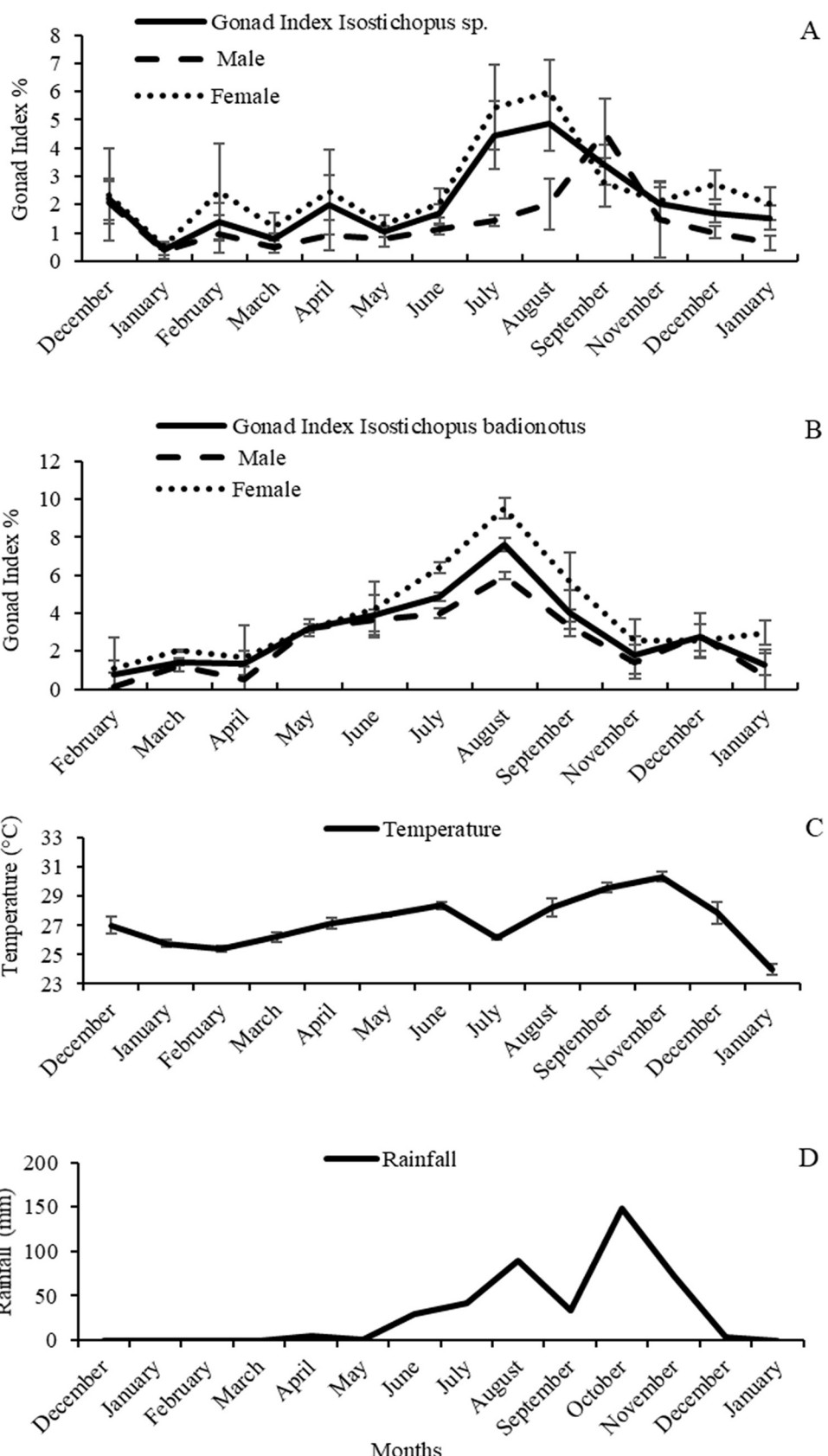

**Fig 3. Average monthly variation of the gonad index and physico-chemical parameters between December 2015 and January 2017.** (A) gonad index *Isostichopus* sp., (B) gonad index *Isostichopus badionotus*, (C) temperature ˚C; (D) rainfall mm. The bars indicate the standard errors.

March a decrease in the percentage of spawned (V), partially spawned (IV) and without gonads (5%, 15% and 45%, respectively) and at the same time individuals in the growth stage (II) (10%) and mature (III) (25%) were recorded, which marked the restart of gametogenesis. From May onwards gonads developed rapidly. In August most gonads reached maturity (76.5% stage III and 5.9% stage II) (Fig 4A), coinciding with the beginning of the rainy season and the increase in temperature (Fig 3C and 3D). Histology revealed that spawning started in September with 7.1% spawned stage (V) and 35.7% partial spawned stage (IV) (Fig 4A). The highest spawning activity occurred between September and November, coinciding with a strong decrease of the GI and the maximum values of temperature and rainfall (Fig 3A, 3C and 3D).

In *I. badionotus*, by February 66.7% of the individuals were partially spawned (IV) and 33.3% mature III. In March, there was a decrease in partially spawned (IV) and mature (III), and at the same time individuals in growth stage (II) and without gonads were observed, marking the restart of gametogenesis (36%, 27%, 9% and 28%, respectively) (Fig 4B). From May onwards, the gonad developed rapidly, reaching a maximum in August with 79% mature (III) and 21% growth (II), coinciding with the maximum value of the GI, an increase in the water temperature and the intensity of rains (Fig 3B, 3C and 3D).

In September, the percentage of mature (III) individuals sharply decreased and partially spawned (IV) increased (21% and 72%, respectively) (Fig 4B), indicating the beginning of the spawning season, which extended until December and was congruent with the decline of the GI and maximum temperature and rainfall records (Fig 3B, 3C and 3D). The spawning season of both *Isostichopus* sp. and *I. badionotus* is characterized by the occurrence of partial spawning and the absence of individuals without gonads.

## 4.7 Fecundity

The average fecundity (± SE) for *Isostichopus* sp. was $16.6 \times 10^6 \pm 2.1 \times 10^6$ oocytes/individual (maximum $56.4 \times 10^6$ and minimum $2 \times 10^6$ oocytes) and in *I. badionotus* was $74.9 \times 10^6 \pm 9.5 \times 10^6$ oocytes/individual (maximum $157.1 \times 10^6$ oocytes and minimum $18.7 \times 10^6$ oocytes). The graphical relationship between the fecundity with the length and the weight of both sea cucumbers is shown in the Supporting information S1 File.

**Table 1. Estimation of the percentage of variation of the Gonad Index (GI) in relation to environmental parameters.**

| Sea cucumbers | Parameter | % annual variation of GI explained by parameter | *n* | Spearman's correlation | *P* | Regression model |
|---|---|---|---|---|---|---|
| *Isostichopus* sp. | Temperature (Temp) | 24.5% | 13 | 0.555 | 0.055 | GI = (-2.14129 + 0.00359296 * temp²) |
| | Salinity (Sal) | 10.2% | 13 | 0.192 | 0.502 | GI = sqrt (-203143 + 0.0222892 * Sal²) |
| | Rainfall (Rain) | 58.6% | 13 | 0.752 | 0.001 | GI = 1.19406 + 0.293588 * sqrt (Rain) |
| *I. badionotus* | Temperature | 42% | 11 | 0.645 | 0.041 | GI = 1/(-2.87434 + 92.7004/temp) |
| | Salinity | 10.7% | 11 | 0.373 | 0.239 | GI = 1/(-1.64506 + 73.7991/sal) |
| | Rainfall | 59.2% | 11 | 0.761 | 0.016 | GI = sqrt (5.23381 + 0.00487448 * rain²) |

**Table 2. Gonad weight of *Isostichopus* sp. and *Isostichopus badionotus*.**

| Sea cucumbers | Gonad weight (g) | | | | | | |
|---|---|---|---|---|---|---|---|
| | Months | All | SE | Female | SE | Male | SE |
| *Isostichopus* sp. | December—2015 | 4.4 | 1.7 | 3.1 | 1.9 | 5.2 | 2.6 |
| | January | 0.7 | 0.3 | 0.9 | 0.3 | 0.6 | 0.4 |
| | February | 2.9 | 1.5 | 5.5 | 3.8 | 1.9 | 1.5 |
| | March | 1.6 | 0.5 | 2.9 | 1.0 | 1.0 | 0.4 |
| | April | 4.6 | 2.6 | 5.7 | 3.7 | 2.1 | 1.3 |
| | May | 2.1 | 0.4 | 2.6 | 0.6 | 1.6 | 0.5 |
| | June | 3.8 | 0.8 | 4.8 | 1.2 | 2.2 | 0.4 |
| | July | 8.9 | 2.3 | 10.9 | 2.8 | 3.0 | 0.9 |
| | August | 7.9 | 1.6 | 9.8 | 1.9 | 3.3 | 1.4 |
| | September | 6.8 | 1.7 | 5.4 | 2.1 | 9.3 | 2.9 |
| | November | 3.3 | 1.0 | 3.4 | 1.1 | 2.2 | 2.2 |
| | December | 3.0 | 0.6 | 4.4 | 1.0 | 2.1 | 0.6 |
| | January—2017 | 2.3 | 0.5 | 2.8 | 0.7 | 1.5 | 0.7 |
| *Isostichopus badionotus* | Months | All | SE | Female | SE | Male | SE |
| | February | 4.9 | 3.5 | 7.0 | 4.8 | 0.6 | 0.0 |
| | March | 10.3 | 3.2 | 14.7 | 0.9 | 8.8 | 4.1 |
| | April | 10.6 | 4.7 | 13.4 | 6.3 | 3.4 | 1.3 |
| | May | 25.5 | 6.5 | 27.2 | 9.5 | 22.3 | 6.6 |
| | June | 23.9 | 2.3 | 23.5 | 4.3 | 24.2 | 3.0 |
| | July | 33.8 | 5.8 | 43.2 | 15.2 | 28.5 | 3.2 |
| | August | 48.4 | 6.3 | 57.1 | 8.9 | 40.9 | 8.5 |
| | September | 28.8 | 4.6 | 32.9 | 8.9 | 27.2 | 5.6 |
| | November | 8.4 | 3.2 | 12.5 | 0.0 | 6.3 | 4.3 |
| | December | 16.3 | 4.0 | 17.6 | 9.6 | 15.6 | 3.9 |
| | January—2017 | 9.3 | 3.2 | 22.3 | 5.7 | 4.9 | 1.5 |

## 5. Discussion

The length and weight of *Isostichopus badionotus* obtained in this study (324 mm and 628 g) were higher than those reported in neighboring countries, such as Venezuela (113 to 474 mm and 58 to 527 g) [23], Panama (329 mm and 214.4 g) [24], Belize (220 mm) [25], Mexico (220 mm and 454 g) [26] and Cuba (612 g) [27]. However, despite the differences observed between regions, the values obtained in this study are within the growth ranges of the species, which can reach a maximum size of up to 450 mm [8, 13].

The recorded size and weight of *Isostichopus* sp. (193 mm and 178 g) were similar to those reported for this same morphotype by Vergara and Rodríguez [28] (189 mm and 232 g) and to that found by Agudelo-Martínez and Rodríguez-Forero [18] (217 g).

However, as the study that identified the genetic differences between *Isostichopus badionotus* and the morphotype *Isostichopus* sp. is recent, we do not have sufficient evidence to affirm or deny that the studies carried out previously in other countries would have taken into account the differences between morphotypes when carrying out their studies.

According to Conand [29] the average weight of most sea cucumbers of the order Aspidochirotida of commercial importance is in the range of 270 to > 400g, *Isostichopus* sp. could be considered as a small species. However, Agudelo-Martínez and Rodríguez-Forero [18] point out that in the Chinese market *I. badionotus* is marketed with a dry weight range of 8–76 g and according to Hernández et al. [30] the conversion rate from wet to dry weight of *Isostichopus*

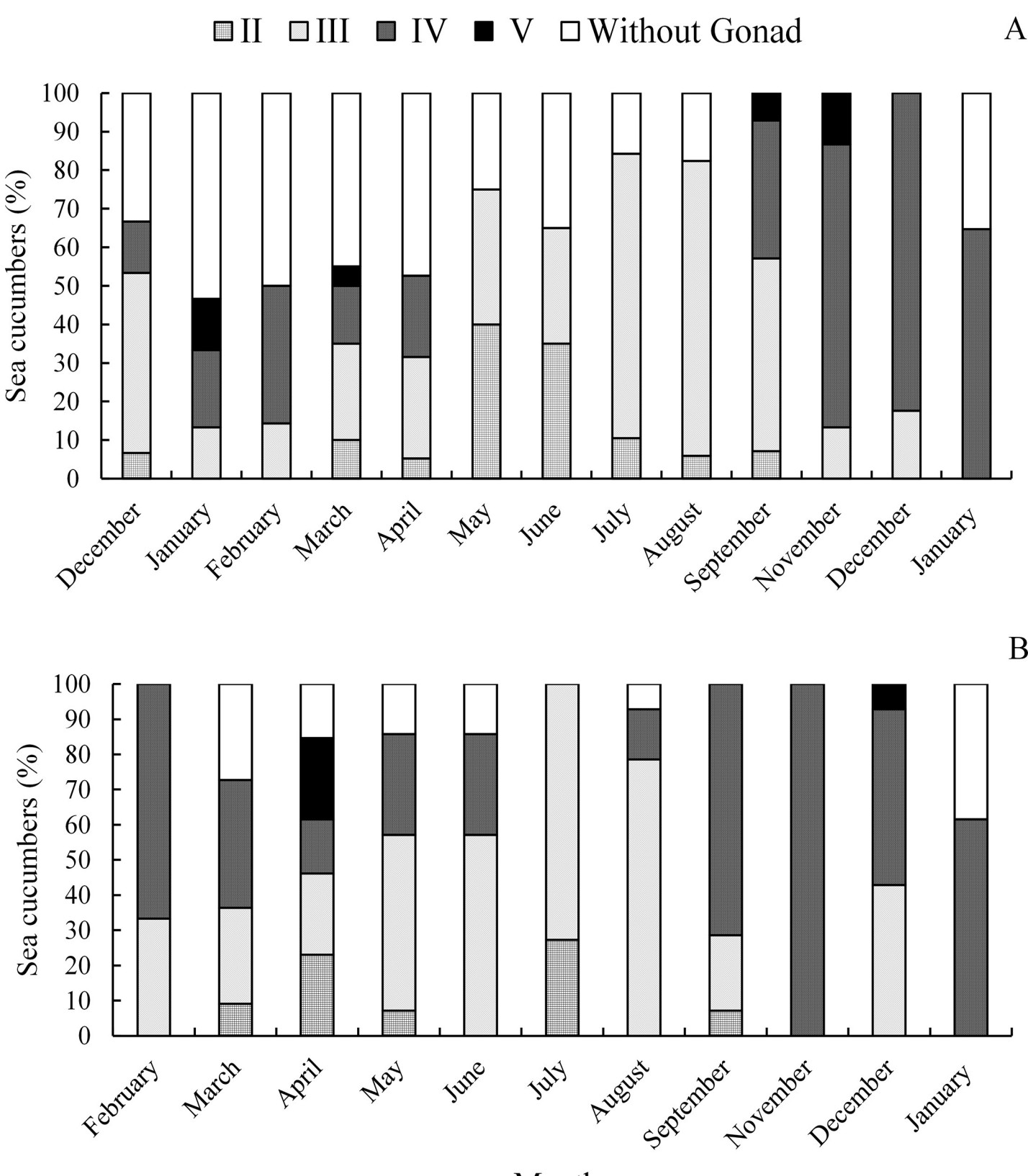

**Fig 4. Stages of gonadal development.** (A) *Isostichopus* sp., (B) *Isostichopus badionotus* between December 2015 and January 2017, (II) growth, (III) mature, (IV) partially spawned and (V) spawned. No stage I individuals were observed in this study (see the text).

sp. is equal to 8% of the body wall weight. Therefore, the estimated dry weight for *Isostichopus* sp. in this study would be 14 g, which is within the marketing range and could in part explain the constant fishing of this sea cucumber [3].

The length and weight distribution frequency recorded in this study are similar to other populations of *I. badionotus* and other holothuroids species, characterized by being unimodal, mostly composed of mature adult individuals with low presence of juveniles [19, 24]. However, it is likely that the juveniles did not have a "low presence" *per se*, but due to their cryptic habits, their size and that they usually occupy places that are difficult to access, would have made observations scarcer.

The occurrence of individuals without gonads in both morphotypes, which were characterized by showing a greater proportion after the spawning season and possessing a smaller size and weight, could be due to a process of reabsorption of the gonad after spawning, which would mainly affect the smaller individuals (young). This assumption is supported by four facts:

1. The total of individuals without gonads possessed all of their internal organs except for the gonad, ruling out the occurrence of auto-evisceration [31].

2. Their average weight and size is greater compared to the weight and size of the smallest individual with gonads in each morphotypes (*I. badionotus*, 210 mm; 362 g and *Isostichopus* sp., 110 mm; 38 g), ruling out that this part of the population is made up entirely of immature individuals.

3. The number of individuals without gonads is greater in the months following spawning, which has also been observed in *I. badionotus* by Lima et al. [32] in Brazil, in *Isostichopus* sp. by Agudelo-Martínez and Rodríguez-Forero [18] in Colombia, and in other sea cucumbers of the order Aspidochirotida by Herrero-Pérezrul et al. [5].

4. The presence in the spawned gonads of both morphotypes of patches composed of dark brown tubules that had intense phagocytosis and are associated with processes of reabsorption from the gonad [6, 7, 33]. In addition, the fact that the number of gonadal tubules increases as the sea cucumber grows [34–36] would suggest that the processes of phagocytosis after spawning would affect the whole gonad in young individuals with a small gonad.

Mean size and weight at first sexual maturity (L50, W50) are parameters commonly used for all fisheries in the world as a reference point for establishing the minimum capture size-weight [7, 29]. The L50 obtained of *I. badionotus* was higher than the one reported for the same species in Panama (130 to 150 mm) and Venezuela (180 mm), and equal to that reported in Cuba (220 mm), countries where strong exploitation of this resource has been reported [20, 37, 38]. However, the values for Panama and Venezuela are similar to the L50 found for *Isostichopus* sp. in this study.

On the other hand, the L50 and W50 obtained in this study show clear differences between the morphotypes: while 50% of the individuals of *Isostichopus* sp. reached the mean size of the first sexual maturity at 175 mm and 155 g, the presence of gonads in *I. badionotus* is only observed in individuals > 210 mm and 362 g. Therefore, both sea cucumbers should have a different management in relation to the minimum size or weight of capture. In the same way, about 50% of the individuals of both morphotypes collected were below the mean size and weight of the first sexual maturity. According to Froese [39], it is a sign of overexploitation when more than 5% of the catches are below the first maturity. However, further studies on the effects of fishing on the population structure, growth and recruitment of these sea cucumbers are needed to support this finding.

Concerning the histological descriptions of the gonad development, both morphotypes showed similar characteristics to those previously described for other species of the order Aspidochirotida [5, 6, 40, 41]. However, the absence of individuals of stage indeterminate (I) may be due to the rapid recovery of the gonad after spawning.

Despite the fact that gonochorism is a common characteristic in holothuroids, the presence of hermaphroditic individuals has been reported in a low percentage in different species, *I. badionotus* 2%, *Holothuria atra* 1.2%, *Isostichopus fuscus* 0.8 and 1.1% [5, 41]. According to Herrero-Pérezrul et al. [42], one of the possible causes of the formation and increase of hermaphroditic individuals is population reduction due to overfishing.

The GI as an indicator of reproductive activity and effort revealed that both *Isostichopus* sp. and *I. badionotus* showed an annual reproductive cycle, where temperature is closely related to gametogenesis and spawning occurs in the warmer months (September–November). These results are consistent with those reported for *I. badionotus* in Panama, Venezuela, and Brazil, where spawning peaks occurred in the warmer months [20, 43].

The influence of temperature on gametogenesis and spawning has been previously reported in several species of tropical holothuroids [5, 22, 35, 40, 44]. However, this study observed rainfall as a further important factor in the gametogenesis and spawning of the sea cucumbers studied. In this regard, Leite-Castro, Souza, Salmito-Vanderley, Nunez, Hamel & Mercier [45] and Benítez-Villalobos, Avila-Poveda & Gutiérrez-Méndez [46] found that during the rainy season there is a significant increase in phytoplankton, related to the contribution of nutrients from runoff and rivers. They also point out that this increased food availability can provide sea cucumbers that ingest phytoplankton (either directly in the case of suspended feeders or as phytodetritus in the case of deposit feeders such as *I. badionotus*) a boost of energy to complete gametogenesis, especially vitellogenesis.

The above corresponds to the observations in the study area, where the Gaira and Manzanares rivers discharge during the rainy season about 280 Ls$^{-1}$ and 330 Ls$^{-1}$ of water loaded with abundant dissolved organic nutrients, respectively [47].

Likewise, it has been reported that several tropical sea cucumbers complete their gametogenesis and have their spawning season during the rainy season, as is the case of *Holothuria mexicana* [20], *H. scabra* [48], *H. grisea* [45] and *H. fuscocinerea* [46]. Benítez-Villalobos et al. [46] point out that sea cucumbers associate their spawning season with rainy periods as a reproductive strategy, because the increase in primary production that is produced would ensure that the larvae are able to feed and survive.

In general, the fecundity calculated for both morphotypes was much higher than that reported for *I. badionotus* by Zacarias-Soto et al. [49] (2 x 10$^5$ to 1.06 x 10$^6$ oocytes/individual) and Palazón [23] (62 x 10$^3$ to 5.06 x 10$^6$ oocytes/individual). Likewise, *I. badionotus* also recorded higher fecundity than species, such as *H. scabra* (9 x 10$^6$ to 17 x 10$^6$ oocytes/individual), *Stichopus variegatus* 7.2 x 10$^6$ to 12.5 x 10$^6$ oocytes/individual) and *Actinopyga mauritiana* (23.6 x 10$^6$ to 33.7 x 10$^6$ oocytes/individual) [50], while *Isostichopus* sp. showed similar values.

The amount of oocytes that can be produced by a sea cucumber is affected by several factors, such as feeding, size or genetics [51] and although the individuals collected in this study showed a larger size and weight than those reported in neighboring countries, it is possible that the method used to calculate fertility has overestimated it.

However, the fact that the fecundity of *I. badionotus* was 3.5 times greater than that of *Isostichopus* sp. could be explained in part by the larger size of *I. badionotus* and in part by the behavior of the morphotypes. On the other hand, *Isostichopus* sp. by showing a more gregarious behavior, sharing shelter and concentrating on specific habitats (the rocky and shallow bottoms) could require a smaller number of gametes during spawning to obtain successful reproduction. While *I. badionotus* by occupying more varied habitats and being more distant

from each other, would require a greater effort in gamete production to increase the chances of reproduction.

Finally, one of the main characteristics observed in both morphotypes during the breeding season was the occurrence of partial spawning. This capacity, that some holothuroids can spawn more than once during the reproductive period, is considered by some researchers as a way to increase the success of reproduction, especially in species that produce very small oocytes [5, 51], as it is the case with *I. badionotus* (100 μm) and *Isostichopus* sp. (98 μm).

## 6. Conclusions

The results of this study indicate that both *Isostichopus badionotus* and *Isostichopus* sp. have an annual reproductive cycle, in which spawning events occur in the warmer months with higher rainfall (September to November). This suggests that warm seawater temperature and /or rainfall act as reproductive synchronizers and spawning triggers in both morphotypes.

In the same way, this study found that *Isostichopus badionotus*, besides showing a significantly larger size and weight than *Isostichopus* sp., had also marked differences in relation to the size and average weight of the first sexual maturity, the fecundity and the depth and type of bottoms they inhabit. These characteristics support the results of Vergara et al. [15], who argue that these are different species, and also imply that the implementation of management and conservation measures (e.g., closures, minimum catch size, allowed fishing gears) should not be the same for both morphotypes.

## Supporting information

**S1 Appendix. Description of the macroscopic and histological characteristics of the ovary and the testes in both *Isostichopus badionotus* and *Isostichopus* sp., throughout the development of gametogenesis.** Including photographs detailing each stage of development.
(DOCX)

**S1 File. Graphic description of the relationship between fecundity and length (S1 Fig, S3 Fig), and fecundity and weight (S2 Fig, S4 Fig) for both *Isostichopus badionotus* and *Isostichopus* sp.**
(DOCX)

## Acknowledgments

We thank The Universidad del Magdalena, especially the Aquaculture Technology Research and Development Group (GIDTA), the Marine and Coastal Research Institute INVEMAR, the Leibniz Centre for Tropical Marine Research (ZMT) GmbH, the Giessen University and Corporation CEMarin, for the support received and the facilities provided for the preparation of this work.

## Author Contributions

**Conceptualization:** Ernesto J. Acosta, Andreas Kunzmann.

**Data curation:** Ernesto J. Acosta.

**Formal analysis:** Ernesto J. Acosta.

**Funding acquisition:** Ernesto J. Acosta, Bernd Werding, Andreas Kunzmann.

**Investigation:** Ernesto J. Acosta.

**Methodology:** Ernesto J. Acosta, Adriana Rodríguez-Forero, Andreas Kunzmann.

**Supervision:** Adriana Rodríguez-Forero, Bernd Werding, Andreas Kunzmann.

**Visualization:** Adriana Rodríguez-Forero, Bernd Werding, Andreas Kunzmann.

**Writing – original draft:** Ernesto J. Acosta.

**Writing – review & editing:** Adriana Rodríguez-Forero, Bernd Werding, Andreas Kunzmann.

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
