## [Decision Letter · Decision Letter 0]

15 Apr 2020

PONE-D-20-04776

Ecological and reproductive characteristics of holothuroids Isostichopus badionotus and Isostichopus sp. native to the Caribbean coast of Colombia.

PLOS ONE

Dear Mr. Acosta Ortiz,

Thank you for submitting your manuscript to PLOS ONE. After careful consideration, we feel that it has merit but does not fully meet PLOS ONE’s publication criteria as it currently stands. Therefore, we invite you to submit a revised version of the manuscript that comprehensively addresses the points raised during the review process.

We would appreciate receiving your revised manuscript by May 30 2020 11:59PM. To enhance the reproducibility of your results, we recommend that if applicable you deposit your laboratory protocols in protocols.io, where a protocol can be assigned its own identifier (DOI) such that it can be cited independently in the future. For instructions see: http://journals.plos.org/plosone/s/submission-guidelines#loc-laboratory-protocols

We look forward to receiving your revised manuscript.

Kind regards,

Michael Schubert

Academic Editor

PLOS ONE

a).    You may seek permission from the original copyright holder of Figure 1 to publish the content specifically under the CC BY 4.0 license.

b).    If you are unable to obtain permission from the original copyright holder to publish these figures under the CC BY 4.0 license or if the copyright holder’s requirements are incompatible with the CC BY 4.0 license, please either i) remove the figure or ii) supply a replacement figure that complies with the CC BY 4.0 license. Please check copyright information on all replacement figures and update the figure caption with source information. If applicable, please specify in the figure caption text when a figure is similar but not identical to the original image and is therefore for illustrative purposes only.

Reviewers' comments:

Reviewer's Responses to Questions

**Comments to the Author**

1. Is the manuscript technically sound, and do the data support the conclusions?

Reviewer #1: Yes

Reviewer #2: Partly

2. Has the statistical analysis been performed appropriately and rigorously? 

Reviewer #1: I Don't Know

Reviewer #2: Yes

3. Have the authors made all data underlying the findings in their manuscript fully available?

Reviewer #1: No

Reviewer #2: Yes

4. Is the manuscript presented in an intelligible fashion and written in standard English?

Reviewer #1: Yes

Reviewer #2: No

5. Review Comments to the Author

Reviewer #1: Please see my appended comments.

It seems that I have to put in 200 total characters into this box in order for the system to accept my review, so I will simply keep typing here until I reach a total of 200 characters.

Reviewer #2: The manuscript describes results of a study about the reproductive periodicity of two species of sea cucumbers found in Columbian waters. Both species are apparently exploited in illegal fisheries. The study has the potential to support more refined management plans for the two species, such as by informing the planning of seasonal closures and minimum legal size limits.

Generally, the manuscript needed more work on tightening the text and editing out typographic flaws prior to submission. Did all of the co-authors carefully read this manuscript and approve of this version to be submitted? This is mandated by the Vancouver Protocol, and indeed specifically by this journal’s policy. Overall, there are numerous careless errors with the writing (and I don’t just mean English usage). Some paragraphs are comprised of one unwieldy sentence, some careless errors with units and precision, flaws in syntax of citing authors. The job of the scientific reviewers is not to edit the manuscripts in this regard. This should be accomplished by the authorship team.

The Results section is far too wordy, and some of it repeats what has already been stated in the Methods (e.g., ANOVA tests and post-hoc tests). I recommend that the authors trim down the Results section by around 40%.

The authors also used a loss-on-ignition method (line 81) to “determine the content of organic matter” in sediments. This method was borrowed from terrestrial ecologists long ago. Bias of this method for analysing tropical sediments that can contain recent carbonates has been long recognised (Dear 1974, Telek and Marshall 1974, Hirota and Szyper 1975, Morse and Mackenzie 1990, Luczak et al. 1997). The problem is that, some carbon will be lost (as CO2) from the mineral component in addition to the oxidative decomposition of organic matter, leading to an overestimation of organic matter. Although this bias is likely to be fairly consistent among treatment and control samples in the current study, the use of such methods in manuscripts like this can be a detriment to the field of research because other authors will consider publication of the work to be peer-approval of the methods for other studies. I believe that the authors must either remove these results from their manuscript (which would not be a great loss anyway) or re-analyse the sediments in a more correct way that does not lead to overestimation of organic matter.

Abstract:

Line 13 says holothurians, but elsewhere it says holothuroids. Be consistent and say holothuroid throughout.

According to the authors, Isostichopus sp. is not yet described taxonomically. Is it indeed a separate species or a putative species?

Line 15 says “the two bays”. But there are not on two bays in this region.

On line 20 and elsewhere in the manuscript, please replace “presented” with something else such as “had”.

You could say “much lower” on line 21.

On line 25, is it the design or the measures that should not be the same for the two species?

Body Text

Syntax or typo errors on lines 22, 63, 69, 70, 82, 84, 103, 120, 125, 138, 144, 152, 177, 191, 214–216, 221, 240, 254, 261, 285, 296, 299, 323, 328–329, 341.

There are many inconsistencies that need to be fixed. For example, sometimes spaces before and after equal sign but sometimes not.

The first paragraph of the Introduction is one big unwieldy sentence—please revise. Same for first part of second para in Introduction. Other instances in the manuscript should be revised too.

Errors with citation styles: authors should not write “described by [15]”, or “”[18] point out”, or “by [30]”. Name the author(s) in such cases. Fix these throughout the manuscript.

Line 50 says “which they named Isostichopus sp.” but the species has not been named.

Don’t start sentences with numbers. Please rewrite them to start with a word.

On line 85, please describe how the animals were measured? Why was a flexible tape measure used? Was the tape contoured to the body such that the length is the contour length? If so, state this.

Describe the ANOVA analyses better. What factors? Are the factors random or fixed? One-way or two-way ANOVA?

All statistical terms should be italicized, including F, p, K, X2, etc.

The precision used for stating values in the Results section is most often unwarranted, especially when one looks at the error estimate. For body weights and lengths, round to nearest whole numbers.

The methods section already says that ANOVA and post-hoc tests were used. So in the Results, there is no need to say things like “ANOVA results showed…”. Likewise, please don’t have statements such as “The percentage….is shown in Fig 4”.

Lines 205–207 says that the GI values of both species were highest in August, with “This increase coincided with an increase in water temperature, the onset of the rainy season and an increase in organic matter concentrations in the sediment.”. Sorry, but from the Figure, I cannot see a good correlation between organic matter and the GI values. These appear unrelated. Please either analyse this correspondence or reword.

The wording in the Discussion could be tightened.

Don’t use hyphens to indicate a range in data or time periods. Use en dashes in these cases. Also, the en dash replaces the word “to” so it is a grammatical error to write “between XX–XX”.

For the Discussion, the authors have not convincingly proven or shown that temperature, rainfall and runoff are all needed for the animals to spawn during late summer. This would imply that if there was a year without much rain, then the animals would not spawn. Instead, be more circumspect and say warm seawater temperature and/or rainfall.

The graph axes going from December 2015 to January 2017 with only one year of months in between looks very strange. I would suggest to just have the months in the axes, and explain about the pooling of monthly data in the captions.

How does the average fecundity of I. badionotus compare with estimates of fecundity of other species? 124 million oocytes seems like a lot. Please can you compare with values from other studies on other species?

6. PLOS authors have the option to publish the peer review history of their article (what does this mean?). If published, this will include your full peer review and any attached files.

Reviewer #1: Yes: Jason Hodin

Reviewer #2: No

---

## [Author Response · Author response to Decision Letter 0]

30 May 2020

We believe that the corrections and suggestions provided by the reviewers were adequate and valuable in improving the manuscript.

The authors adjusted the manuscript as best as possible to the requirements and suggestions provided by the reviewers and the editor.

---

## [Decision Letter · Decision Letter 1]

13 Jul 2020

PONE-D-20-04776R1

Ecological and reproductive characteristics of holothuroids Isostichopus badionotus and Isostichopus sp. native to the Caribbean coast of Colombia.

PLOS ONE

Dear Dr. Acosta Ortiz,

Thank you for submitting your manuscript to PLOS ONE. After careful consideration, we feel that it has merit but does not fully meet PLOS ONE’s publication criteria as it currently stands. Therefore, we invite you to submit a revised version of the manuscript that comprehensively addresses the points raised during the review process.

It is requested that the response letter very carefully details the changes made in response to the reviewers' comments and that it includes clear justifications as to why certain comments have been implemented in the revised manuscript. Furthermore, given the lack of quality of the manuscript text, the writing will have to be improved significantly to render the manuscript acceptable in PLOS ONE. The measures taken to improve the manuscript text need to be indicated in the response letter.

We look forward to receiving your revised manuscript.

Kind regards,

Michael Schubert

Academic Editor

PLOS ONE

Reviewers' comments:

Reviewer's Responses to Questions

**Comments to the Author**

1. If the authors have adequately addressed your comments raised in a previous round of review and you feel that this manuscript is now acceptable for publication, you may indicate that here to bypass the “Comments to the Author” section, enter your conflict of interest statement in the “Confidential to Editor” section, and submit your "Accept" recommendation.

Reviewer #1: (No Response)

Reviewer #2: (No Response)

2. Is the manuscript technically sound, and do the data support the conclusions?

Reviewer #1: Yes

Reviewer #2: No

3. Has the statistical analysis been performed appropriately and rigorously? 

Reviewer #1: Yes

Reviewer #2: Yes

4. Have the authors made all data underlying the findings in their manuscript fully available?

Reviewer #1: Yes

Reviewer #2: Yes

5. Is the manuscript presented in an intelligible fashion and written in standard English?

Reviewer #1: Yes

Reviewer #2: No

6. Review Comments to the Author

Reviewer #1: I would like to congratulate the authors on their substantial efforts to revise the manuscript. It is now much more readable and the new analyses have, in my opinion, greatly improved the manuscript. I only have a few minor suggestions remaining.

If the authors revise the manuscript in the ways I suggest here, I would then deem it suitable for publication. If the authors object to any of these changes, please state their reasons why.

I have also pointed out a few times below where I would like the authors to include -in the manuscript itself- issues that they wrote about in their response to my original review, yet did not mention in the revised manuscript itself, as far as I can tell.

Below, I reference the line numbers in the new version of the paper.

=====

Lines 20-22: please change to read:

“Isostichopus sp. had an average size and weight (193 + 52 mm and 178 + 69 g) and size and weight at first maturity (175 mm and 155 g) that was much lower than I. badionotus (respectively, 324 + 70 mm and 628 + 179 g; 220 mm and 348 g).”

Line 76: change “divers’ to “snorkelers”

Line 78: This would be a good place to mention what the authors mentioned in their response to my original review: that there are likely individuals below 10 meters, but this could not be assessed in the current study.

Lines 89-94 should be a single sentence. Therefore, please change Lines 90-92 to read:

“…using a flexible 1-m tape measure at the nearest 0.5 cm (this tape contoured the body, so that the total length is the contour length); gonad volume using

Line 106: change “states” to “stages”

Line 134: please give version of R software used

Line 153 and 155: change “had not gonads” to “did not have gonads”

Lines 164, 290, 292, 294, 297, 299, 305: Thank you for providing p-values. But please do not provide that many figures after the decimal place. 3 is sufficient.

So, for example, on line 164 please either write p=0.039 or something like p<0.04.

The K values on lines 164 and 304 are probably also a bit too precise. e.g., on line 164, “6.513” is probably sufficient.

Line 171: Please change “with” to “that was”

Lines 189, 199 and elsewhere if relevant: When listing the F statistics please subscript the degrees of freedom numbers. So, on line 189, the “2,111” should appear as a subscript following “F”

Lines 189, 199, 304: p=0.0 is not acceptable. Please write p<0.001

Lines 228, 258: Change “micras” to “µm”

Line 247: change “testicles” to “testes”

Line 310: In Table 1, the first row of data is missing cell borders.

Line 331: please add to figure legend: “No Stage I individuals were observed in this study (see the text).”

Line 333: change “whose presence” to “such individuals”

Line 353-356: I still have a problem with the way the exponent figures are listed here. It is very random. Can the authors please make it more easily comparable between species? I suggest rewriting all of these figures as x 10^6

Line 380: Can the authors please add (perhaps here) a note regarding what they wrote in their response to my original review? Namely, that it’s not likely that the individuals have “low presence” per se, but that due to their size and cryptic habitat, few were observed.

Line 392: please fix this reference where it says “Aspidochrotida [5] and 4”

Lines 419-421. interesting. is it possible that hermaphroditism increasing in exploited areas is also possibly due to other co-occurring environmental inputs? In areas with heavy fishing there may also be correlated heavy run-off of land-based pollutants.

Line 472: please correct the word “arts” here

Somewhere in discussion: can the authors please somewhere point out what they wrote in their response to my original review, namely:

“we do not have sufficient evidence to affirm or deny that the studies carried out previously in other countries would have made a correct identification of the species in the light of the new findings.”

I believe this is important to mention as it is relevant to their comparisons with prior studies that they made throughout the paper.

Figs:

Fig 2 & 5: please make sure that the quality of Figs 2 & 5 are improved when published. The resolution in my version was poor.

Fig 3: panels D,H,L & P still need scale bars

Fig 6: I’m afraid I still can’t distinguish the grey colors of stages II and III. Can the authors please try one more time to make a visibly different set of grey scale colors for stages II-V in Fig. 6?

Thank you for including the fecundity x body size plots in you response to my original review. I strongly suggest that the authors include these four plots as supplementary material.

Reviewer #2: In the title, just write “in Colombia” instead of “native to the Caribbean coast of Colombia”.

Authors, simply writing over and over in the response document “Has been changed accordingly to the recommended” does not tell the reviewer or Editor where to find the correction in the revised file, or what was done in some instances. In doing so, you are making the job harder for the reviewer.

Subscript the degrees of freedom after the F when reporting F-values.

Okay for the removal of the analysis of organic matter from the manuscript.

Could the authors please double-check all of their references to Figures? For example, line 283 states “an increase in water temperature and the onset of the rainy season ( Fig 5A, B, C)” but that should then be Fig. 5C, D.

To the first submission, I advised “On line 20 and elsewhere in the manuscript, please replace “presented” with something else such as “had”.” The authors replied “Answer: Has been changed accordingly to the recommended”, yet this remains a grammatical flaw in a couple places in the manuscript.

You cannot have p = 0.0 in an ecological study such as this. Please fix this (surely the senior authors can help out here).

Authors were asked in the first review to revise instances in the manuscript where sentences were long and unwieldy. They claim in the response document to have changed this, but there are still clear examples in the manuscript. Just as a glaring example, the second sentence of the Introduction is 68 words long! This is not right for a reviewer to have to make the same recommendation twice.

Authors were asked in the first review to fix up the precision used for stating values in the Results section. Again, they claim in the response document to have changed this, but there is still unnecessary precision in the manuscript. For example, lines 146, 147, 150, 152-155, 164, 168, 169 and so on…. Again, this is not right for a reviewer to have to make the same recommendation twice.

In the first review, I noted the specific lines with typos or grammatical errors. Many were not fixed up. The authors need to go back over the original submission and fix the errors on those lines.

The process of contouring the measuring tape to the body to measure it’s length is not a standard practice for any other animal I can think of. This seems to be a methodological flaw.

In the first review, I noted “Don’t use hyphens to indicate a range in data or time periods. Use en dashes in these cases.”. This still appears as an error in some cases.

Still typos. E.g., line 392 “sea cucumbers of the order Aspidochirotida [5] and 4.”. Line 258 “The bar indicates 100 micras”. Please, all authors, proof-read the version before submission. You are all supposed to approve the submission.

The Results section in the original submission was 151 lines. I advised that it was too long and wordy, and recommended to shorten it by around 40%. Instead, the authors have expanded this section and it is now 222 lines. Sorry, this section remains too wordy and I cannot offer my recommendation to have it published like this. So many of the sentences are verbose; there are words that are unnecessary. There are four authors on this manuscript, some of them well published. I am sure some of them could help to revise this section, even if the lead author is a doctoral student. Moreover, it is their duty to do so as authors, according to this journal’s ethics policy, not to mention international codes of ethics including COPE and the Vancouver Protocol.

I like this study. But I am still concerned about the fact that the authors state throughout the manuscript that Isostichopus sp. is a separate species, yet there is no rigorous scientific results to show this. Their response document says only that “We believe they are two separate species.”, but science must be founded on more than beliefs. Without genetic or other morphological evidence to show that they are two distinct species, this should not be reported as such in this manuscript. I am not intending to be difficult here. Rather, this just appears incongruent. Perhaps the authors could reconsider the way in which this putative species is framed in this manuscript?

In the first review, I gave several recommendations and asked for clarification in the Discussion. The author simply replied “Has been changed accordingly to the recommended”, but I cannot see what changes were made and how these points have been addressed. This approach for responding to careful comments by reviewers is grossly insufficient.

7. PLOS authors have the option to publish the peer review history of their article (what does this mean?). If published, this will include your full peer review and any attached files.

Reviewer #1: **Yes: **Jason Hodin

Reviewer #2: No

---

## [Author Response · Author response to Decision Letter 1]

29 Aug 2020

Comments from the editor and reviewers have been responded to in detail in the response letter.

---

## [Editor Report · Decision Letter 2]

9 Sep 2020

PONE-D-20-04776R2

Ecological and reproductive characteristics of holothuroids Isostichopus badionotus and Isostichopus sp. in Colombia

PLOS ONE

Dear Dr. Acosta Ortiz,

Thank you for re-submitting your manuscript to PLOS ONE. Following an evaluation of the submission and discussions with the reviewers, I am hereby returning the manuscript, granting you one, final, opportunity to comprehensively address the shortcomings of the manuscript. In the last decision letter sent to you, I specifically requested "that the response letter very carefully details the changes made in response to the reviewers' comments and that it includes clear justifications as to why certain comments have been implemented in the revised manuscript. Furthermore, given the lack of quality of the manuscript text, the writing will have to be improved significantly to render the manuscript acceptable in PLOS ONE. The measures taken to improve the manuscript text need to be indicated in the response letter."

Yet, the revised manuscript has not been amended according to the reviewers' comments and the manuscript text has not been revised extensively. Although, based on the PLOS ONE publication criteria, these shortcomings would justify rejection of the manuscript in its current form, I am willing to reconsider a revised manuscript, in which all comments have been addressed comprehensively.

Should you decide to re-submit your manuscript, you will need to include a very detailed description of how the manuscript was improved. In addition, you will need to include a signed statement from each one of the authors detailing their respective contribution to the study, the writing of the manuscript and its revision. This measure is necessary, as the reviewers have noted potential ethical issues related to a possible lack of involvement of some of the authors in the compilation and revision of the work.

A detailed rebuttal letter that responds to each point raised by the academic editor and reviewer(s). You should upload this letter as a separate file labeled 'Response to Reviewers'.Individual statements from the authors of the study detailing their respective contribution to the study, the writing of the manuscript and its revision.A marked-up copy of your manuscript that highlights changes made to the original version. You should upload this as a separate file labeled 'Revised Manuscript with Track Changes'.An unmarked version of your revised paper without tracked changes. You should upload this as a separate file labeled 'Manuscript'.

We look forward to receiving your revised manuscript.

Kind regards,

Michael Schubert

Academic Editor

PLOS ONE

---

## [Author Response · Author response to Decision Letter 2]

27 Nov 2020

In this new version of the manuscript, the text was reduced throughout the manuscript, unnecessary words, phrases and repetitive or redundant words were deleted. With special emphasis on the results section, which went from 2106 words in the first version to 1625 in this version.

Also the concept of separate species was redefined throughout the manuscript, indicating that they are two morphotypes of the same species.

In the discussion the effect of rainfall on gonadal development and spawning has been addressed in more depth. Based on references we explain the influence that the rainy season has on the increase of primary production and consequently the development of gonads in sea cucumbers.

Finally, in this new version of the manuscript, numerous editorial corrections were made (spelling, grammar and syntax corrections) in order to improve the quality of the manuscript.

All changes that were made are detailed in the letter of response to comments from reviewers, indicating the line number of the change in the manuscript with change tracking.

---

## [Decision Letter · Decision Letter 3]

4 Jan 2021

PONE-D-20-04776R3

Ecological and reproductive characteristics of holothuroids Isostichopus badionotus and Isostichopus sp. in Colombia

PLOS ONE

Dear Dr. Acosta Ortiz,

Thank you for submitting your manuscript to PLOS ONE. After careful consideration, we feel that it has merit but does not fully meet PLOS ONE’s publication criteria as it currently stands. Therefore, we invite you to submit a revised version of the manuscript that addresses the points raised during the review process.

We look forward to receiving your revised manuscript.

Kind regards,

Michael Schubert

Academic Editor

PLOS ONE

Reviewers' comments:

Reviewer's Responses to Questions

**Comments to the Author**

1. If the authors have adequately addressed your comments raised in a previous round of review and you feel that this manuscript is now acceptable for publication, you may indicate that here to bypass the “Comments to the Author” section, enter your conflict of interest statement in the “Confidential to Editor” section, and submit your "Accept" recommendation.

Reviewer #1: (No Response)

2. Is the manuscript technically sound, and do the data support the conclusions?

Reviewer #1: Yes

3. Has the statistical analysis been performed appropriately and rigorously? 

Reviewer #1: Yes

4. Have the authors made all data underlying the findings in their manuscript fully available?

Reviewer #1: (No Response)

5. Is the manuscript presented in an intelligible fashion and written in standard English?

Reviewer #1: Yes

6. Review Comments to the Author

Reviewer #1: I am pleased that the authors have been given a final chance to revise the manuscript and that they have done so with appropriately detailed responses to my and the other reviewer's comments.

I have very few reemaining comments, which would need to be addressed before publication, but they are all quite minor. Please see the attached comments

7. PLOS authors have the option to publish the peer review history of their article (what does this mean?). If published, this will include your full peer review and any attached files.

Reviewer #1: **Yes: **Jason Hodin

---

## [Author Response · Author response to Decision Letter 3]

30 Jan 2021

All the recommendations suggested by the reviewers have been taken into account and the respective corrections have been made in the manuscript. Please note that the new line numbers refer to the manuscript with Track Changes. 

The changes described below have also been described in the response letter to the reviewers.

1. BODY MEASUREMENTS 

This was what I wrote in my 2nd review: 

Lines 89-94 should be a single sentence. Therefore, please change Lines 90-92 to read: “...using a flexible 1-m tape measure at the nearest 0.5 cm (this tape contoured the body, so that the total length is the contour length); gonad volume using…” 

The authors wrote this in their latest response:

Answer: Has been changed accordingly. The changes can be seen in the manuscript with corrections in the lines: 108 – 113. 

But the authors did not include the parenthetical I requested regarding contouring the body with the flexible tape. I will note that Reviewer 2 had a similar comment: 

REV 2: "The process of contouring the measuring tape to the body to measure it’s length is not a standard practice for any other animal I can think of. This seems to be a methodological flaw." To which the authors responded in detail: 

Answer: The measurement was made on the ventral part of the sea cucumber, which is the flattest part of the animal. Measuring from the mouth to the anus. However, due to the ability of these animals to elongate and contract at will, it was decided to standardize the measurements, taking them just after sacrifice by thermal shock in water at 4°C, as this guaranteed that all the animals were in a state of contraction, thus reducing the error. However, in some cases the contracted individuals had a C or S shape for which a flexible metric measuring tape was used to follow the C or S-shape contour of the body. We have deleted this misleading sentence now. 

This is not an acceptable solution., As Reviewer 2 notes, the measurement technique employed was unorthodox. Here, the authors have justified that unorthodox technique, but something like the above description (i.e., of the thermal shock treatment preceding measurement, and thus necessitating contour measurements) needs to be included in the Methods. Please add this text before publication.

New answer: The detailed description of the method used for the measurement of sea cucumbers has now been included in the methodology. The changes can be seen in the manuscript with corrections in the lines: 95 – 100.

2. FIGURE EXPONENTS

This was what I wrote in my 2nd review:

 Line 353-356: I still have a problem with the way the exponent figures are listed here. It is very random. Can the authors please make it more easily comparable between species? I suggest rewriting all of these figures as x 10^6.

The authors wrote this in their latest response:

Answer: Has been corrected. The changes can be seen in the manuscript with corrections in the lines: 429 – 432.

Thank you. But the authors appeared to have made an error when changing the figure exponents. Here is how the prior version read:

I. badionotus, the estimated average fecundity (+ SE) was 74.9 x 10^7 + 9.5 x 10^7.

 ...and here is how the current version reads (Lines 266-267).

I. badionotus, the estimated average fecundity (+ SE) was 74.9 x 10^6 + 9.5 x 10^6.

The authors seemed to have forgotten to change the values preceding the exponents here as the correct value would appear to be 749 x 10^6 + 95 x 10^6.

 Please correct this before publication.

New answer: The average fecundity figures for I. badionotus are now correct and agree with the values in the graph of the supplementary material S1 Fig.

The confusion was due to a typing error in the first version submitted, where it was written that I. badionotus had a fertility of 74.9 x 10^7 + 9.5 x 10^7, while it really was 74.9 x 10^6 + 9.5 x 10^6.

3. LINE 159-160 Please change this sentence to read as follows: The sex ratio was not significantly different from 1:1 (I. badionotus: n = 100; X2 = 1.4; p > 0.05; Isostichopus sp.: n = 158; X2 = 3.3; p > 0.05).

New answer: Has been changed accordingly. The changes can be seen in the manuscript with corrections in the lines: 163 – 164. 

4. LINE 273 Please change this sentence to read as follows: ...higher than those reported in neighboring countries. 

New Answer: Done, please see corrections in the line 277.

5. LINE 301 AND FOLLOWING

This assumption is supported by four facts....

Both I and Rev 2 had problems with this section. I think the problem is that the "four facts" is written as one extremely long sentence.

I would like to see these four facts separated as a numbered list. I.e.,

This assumption is supported by four facts:

1) The total of individuals without gonads possessed all of their internal organs except for the gonad, ruling out the occurrence of auto-evisceration [31]; 

2) Their average weight and size is greater compared to the weight and size of the smallest individual with gonads...

And so on. If the section is formatted this way (as a separated numbered list) then readers will have a much easier time digesting this information. 

New Answer: Has been changed accordingly. The changes can be seen in the manuscript with corrections in the lines 305 - 316.

6. LINE 384-385

Please change this sentence to read as follows:

...especially in species that produce very small oocytes [5, 51], as it is the case with I. badionotus (100 μm) and Isostichopus sp. (98 μm).

New answer: Has been changed accordingly. The changes can be seen in the manuscript with corrections in the line 391.

7. SUPPL S1 Fig (1).docx Thank you for including these figures as supplementary material, but when I opened this attachment, I only saw the Figure captions/legend. The four Figures themselves were not present! Please fix this before publication!!

New answers: The inclusion of the figures has now been verified.

---

## [Editor Report · Decision Letter 4]

3 Feb 2021

Ecological and reproductive characteristics of holothuroids Isostichopus badionotus and Isostichopus sp. in Colombia

PONE-D-20-04776R4

Dear Dr. Acosta Ortiz,

We’re pleased to inform you that your manuscript has been judged scientifically suitable for publication and will be formally accepted for publication once it meets all outstanding technical requirements.

Kind regards,

Michael Schubert

Academic Editor

PLOS ONE

---

## [Editor Report · Acceptance letter]

12 Feb 2021

PONE-D-20-04776R4 

Ecological and reproductive characteristics of holothuroids *Isostichopus badionotus* and *Isostichopus* sp. in Colombia 

Dear Dr. Acosta:

I'm pleased to inform you that your manuscript has been deemed suitable for publication in PLOS ONE. Congratulations! Your manuscript is now with our production department. 

Kind regards, 

on behalf of

Dr. Michael Schubert 

Academic Editor

PLOS ONE